# Parallel evolution between genomic segments of seasonal human influenza viruses reveals RNA-RNA relationships

**Jennifer E Jones**[1,2,3], **Valerie Le Sage**[1,3], **Gabriella H Padovani**[1,3], **Michael Calderon**[4], **Erik S Wright**[2,5]*, **Seema S Lakdawala**[1,3]*

[1]Department of Microbiology & Molecular Genetics, University of Pittsburgh, Pittsburgh, United States; [2]Center for Evolutionary Biology and Medicine, University of Pittsburgh, Pittsburgh, United States; [3]Center for Vaccine Research, University of Pittsburgh School of Medicine, Pittsburgh, United States; [4]Department of Cell Biology, Center for Biologic Imaging, University of Pittsburgh, Pittsburgh, United States; [5]Department of Biomedical Informatics, University of Pittsburgh, Pittsburgh, United States

**Abstract** The influenza A virus (IAV) genome consists of eight negative-sense viral RNA (vRNA) segments that are selectively assembled into progeny virus particles through RNA-RNA interactions. To explore putative intersegmental RNA-RNA relationships, we quantified similarity between phylogenetic trees comprising each vRNA segment from seasonal human IAV. Intersegmental tree similarity differed between subtype and lineage. While intersegmental relationships were largely conserved over time in H3N2 viruses, they diverged in H1N1 strains isolated before and after the 2009 pandemic. Surprisingly, intersegmental relationships were not driven solely by protein sequence, suggesting that IAV evolution could also be driven by RNA-RNA interactions. Finally, we used confocal microscopy to determine that colocalization of highly coevolved vRNA segments is enriched over other assembly intermediates at the nuclear periphery during productive viral infection. This study illustrates how putative RNA interactions underlying selective assembly of IAV can be interrogated with phylogenetics.

*For correspondence:
eswright@pitt.edu (ESW);
lakdawala@pitt.edu (SSL)

**Competing interest:** The authors declare that no competing interests exist.

## Introduction

Inordinately high genetic variation is a hallmark of RNA viruses. The rapid evolution underlying this variation can occur as a result of mutation, recombination, or reassortment, with major consequences for human disease (*Andino and Domingo, 2015*). In the case of influenza virus, these consequences include poor vaccine efficacy rates, immune escape, antiviral resistance, and the emergence of novel strains (*Lyons and Lauring, 2018*). Within the past century, influenza A virus (IAV) pandemics occurred in 1918 (H1N1), 1957 (H2N2), 1968 (H3N2), and 2009 (H1N1) (*Neumann et al., 2009*; *Paules and Subbarao, 2017*; *Short et al., 2018*). Each of the last three influenza pandemics was attributable to a reassortant strain composed of a novel combination of the eight viral RNA (vRNA) segments of the influenza virus genome (*Neumann et al., 2009*). Thus, the emergence of pandemic strains is marked by a concomitant alteration in the influenza virus genome.

Public health measures to limit the impact of influenza virus outbreaks prioritize emerging viruses based on perceived risk factors such as the potential for reassortment between circulating influenza viruses. Reassortment of vRNA segments must occur during selective assembly of all eight genomic segments, which occurs after export of vRNA segments from the nucleus (*Lakdawala et al., 2014*). Genomic assembly contributes to heterogeneity in progeny viruses and could determine the fitness

**eLife digest** The viruses responsible for influenza evolve rapidly during infection. Changes typically emerge in two key ways: through random mutations in the genetic sequence of the virus, or by reassortment. Reassortment can occur when two or more strains infect the same cell. Once in a cell, viral particles 'open up' to release their genetic material so it can make copies of itself using the cell's machinery. The new copies of the genetic material of the virus are used to make new viral particles, which then envelop the genetic material and are released from the cell to infect other cells. If several strains of a virus infect the same cell, a new viral particle may pick up genetic segments from each of the infecting strains, creating a new strain via reassortment.

Several factors are known to affect the success of the reassortment process. For example, if the new strain acquires a genetic defect that hinders its replication cycle, it is likely to die out quickly. Other times, this trading of genetic information can create a strain that is more resistant to the human immune system, allowing it to sweep across the globe and cause a deadly pandemic. However, a key part of the reassortment process that still remains unclear is how genome segments from two different influenza strains recognize each other before merging together to create hybrid daughter viruses.

To explore this further, Jones et al. used a technique called fluorescence microscopy. They found that genome segments that evolved along similar paths were more likely to cluster in the same area inside infected cells, and therefore, more likely to be reassorted together into a new strain during assembly of daughter viruses. This suggests that assembly may guide the evolutionary path taken by individual genomic segments. Jones et al. also looked at the evolution of different genome segments collected from patients suffering from seasonal influenza, and found that these segments had a distinct evolutionary path to those in pandemic-causing strains.

This research provides new insights into the role of reassortment in the evolution of influenza viruses during infection. In particular, it suggests that how the genome segments interact with one another may have a previously unknown and important role in guiding this evolution. These insights could be used to predict future reassortment events based on evolutionary relationships between influenza virus genomic segments, and may in the future be used as part of risk assessment tools to predict the emergence of new pandemic strains.

of reassortant strains after coinfection (*Brooke, 2017*; *Lowen, 2017*). Selective assembly is thought to be facilitated by intersegmental RNA-RNA interactions. Each vRNA segment encodes packaging signals that must be compatible for reassortment to occur (*Lowen, 2017*; *Richard et al., 2018*). Although much remains unknown about the role of RNA-RNA interactions in genomic assembly, it is evident that disruption of interactions between two vRNA segments can alter interactions with other segments, leading to a model in which hierarchical interactions between vRNA segments ensure selective assembly (*Dadonaite et al., 2019*; *Le Sage et al., 2020*; *Marsh et al., 2008*). Such complexity among vRNA interactions poses a significant hurdle to reassortment (*Gavazzi et al., 2013*; *Noda et al., 2006*). It is consequently imperative to identify the evolutionary constraints imposed by intersegmental vRNA interactions, as this may improve risk assessment efforts for emerging influenza viruses.

Complex intersegmental RNA-RNA interactions could be governed by epistasis, the phenomenon by which a mutation in one gene is impacted by the presence or absence of mutations in other genes (*Sardi and Gasch, 2018*). A number of tools exist to examine the shared evolutionary trajectories resulting from epistatic interactions between genes, yet the current focus surrounds constraints on indirect interactions between proteins that may function together rather than on interactions between viral RNA (*Escalera-Zamudio et al., 2020*). Previous work with probabilistic models revealed that several mutations in the influenza virus nucleoprotein (NP) that are destabilizing on their own became fixed as a result of counterbalancing compensatory mutations that improve the overall protein stability of NP (*Gong et al., 2013*). These destabilizing mutations occur within T cell epitopes of NP that may be important for immune escape (*Gong et al., 2013*). Stabilizing epistasis was similarly instrumental to the emergence of oseltamivir resistance mutations in the influenza neuraminidase (NA) (*Bloom et al., 2010*). The rise of oseltamivir resistance mutations in NA spurred investigation of shared evolutionary trajectories, or parallel evolution, between NA and hemagglutinin (HA), demonstrating that

mutations in HA may have facilitated acquisition of oseltamivir resistance mutations in NA (*Jang and Bae, 2018*; *Kryazhimskiy et al., 2011*; *Neverov et al., 2015*). We propose that phylogenetics could be further employed to investigate epistasis arising from direct RNA-RNA interactions between IAV segments. Therefore, shared evolutionary trajectories, or parallel evolution, between vRNA segments could reveal epistatic constraints on genomic assembly and reassortment.

In this study, we set out to combine phylogenetics and molecular biology to examine parallel evolution across vRNA segments genome-wide in seasonal human influenza viruses to identify potential epistatic relationships. Unlike previous studies, our objective was to identify relationships between vRNA segments that might play key roles specifically in genomic assembly. To evaluate phylogenetic relationships among vRNA segments, we relied upon the Robinson-Foulds distance (RF), a widely used measure of topological distance between trees (*Robinson and Foulds, 1981*). This method determines the number of branch partitions that are not shared between two trees (*Robinson and Foulds, 1981*) and is therefore a quantitative measure of the topological distance between trees. We combined the conventional RF with the clustering information distance (CID), a recently described measure of tree similarity with greater sensitivity for distinguishing between trees (*Smith, 2020*). Lower RF/CID corresponds with greater tree similarity, with a tree distance of 0 indicating that two trees are topologically equivalent. Our approach relies upon the assumption that tree distance would be inversely correlated with the degree of parallel evolution between genome segments arising from either RNA-RNA or protein-protein interactions. Incompatible polymerase subunits exhibit replication deficiencies and are known restriction factors in reassortment (*Li et al., 2008*). Accordingly, we would predict that trees built from PB2, PB1, and PA would have high similarity, reflective of a shared evolutionary trajectory. Likewise, mounting evidence from our group and others suggests that direct intermolecular interactions between vRNA segments coordinate selective assembly (*Dadonaite et al., 2019*; *Le Sage et al., 2020*). Highly similar trees could therefore also be reflective of direct interactions between vRNA segments that may facilitate selective packaging. To distinguish between the roles of RNA and protein, we further examine tree similarity in viral proteins, choosing gene segments with high gene tree similarity, but not high protein tree similarity, to probe for RNA-RNA interactions. Since genomic assembly occurs in the cytoplasm after nuclear export (*Lakdawala et al., 2014*), we reasoned that assembly intermediates found in close proximity to the nucleus could serve as scaffolds for genomic assembly and sought to visualize this by confocal microscopy. Therefore, our approach systematically identifies putative epistatic relationships between vRNA segments to elucidate mechanisms of selective vRNA assembly.

## Results

### Tree similarity between vRNA segments is not uniform in H3N2 viruses

H1N1 and H3N2 viruses have cocirculated in the human population since 1977 (*Neumann et al., 2009*). In order to identify shared evolutionary trajectories between vRNA segments in seasonal human IAV strains over time, we examined parallel evolution between vRNA segments in viruses representative of each subtype from multiple time periods (*Table 1*). Bracketing H3N2 viruses into two time intervals permitted investigation of conserved vRNA relationships over time in antigenically drifted H3N2 viruses. We took a similar approach with human H1N1 viruses, bracketing instead on the antigenic shift event in 2009 and the emergence of the pandemic swine-origin H1N1 virus in

**Table 1.** Influenza A virus strain datasets.

Human H1N1 or H3N2 virus sequences for which full-length sequences are available (Influenza Research Database). Representative sequences were selected for further analyses by clustering. 'Final clusters' indicates the number of clusters after small clusters were collapsed or omitted.

| Subtype | Time period | Total strains | Clusters with >97% identity | Final clusters |
|---------|-------------|---------------|------------------------------|----------------|
| H3N2 | 1995–2004 | 1026 | 16 | 12 |
| | 2005–2014 | 3879 | 17 | 12 |
| H1N1 | 2000–2008 | 821 | 11 | 9 |
| | 2010–2018 | 4072 | 14 | 9 |

the human population. Comparison of vRNA relationships in pre-pandemic (2000–2008) and post-pandemic (2010–2018) human H1N1 viruses could reveal distinct vRNA relationships from viruses of two distinct lineages or alternatively, uncover vRNA relationships that remain conserved despite swapping of vRNA segments across multiple host species.

Our approach outlined in *Figure 1* examines evolutionary relationships between vRNA segments. We began our investigation with all seasonal human H3N2 viruses for which full-length sequence information was available in the Influenza Research Database (IRD), yielding 1026 H3N2 viruses from 1995 to 2004 and 3879 H3N2 viruses from 2005 to 2014 (*Table 1*). Reconstructing phylogenetic trees from all available sequences was disadvantageous, as a preliminary analysis of 300 sequences suggested that a great deal of phylogenetic variation could not be statistically supported by bootstrapping (branch support less than 70). This lack of bootstrap support was problematic for our downstream analysis of tree similarity, since topological distance can result from misleading phylogenetic signal when branches are poorly supported. Thus, reliance upon larger, poorly resolved trees would lead to uninterpretable tree distances. To address this, we used a clustering approach to select representative strains that would produce more statistically robust trees. We first concatenated sequences from all strains into full-length genomes from which we built alignments (*Figure 1A*) and clustered into operational taxonomic units on a neighbor-joining species tree (*Figure 1B*). Despite the fact that fewer full-length influenza virus genomic sequences were available prior to the 2000 s, our approach resulted in a similar number of clusters within a subtype (*Table 1*), consistent with the notion that increased sequencing has led to more closely related sequences in public databases.

The primary objective behind clustering was to minimize variation between trees that was not statistically supported by bootstrapping. The cutoff for sequence identity during clustering of the species tree was therefore an important consideration because it controlled how much unsupported variation remained in our trees. Higher cutoffs (98–99% sequence identity) yielded species trees with more clusters containing fewer members while lower cutoffs (95–96% sequence identity) contained increasingly fewer clusters with more members grouped in each cluster. We selected a cutoff of 97 % sequence identity based on the observation that it produced vRNA trees with an intermediate number of clusters (16–17 clusters in each species tree). We selected several high-quality sequences from each cluster to build replicate vRNA trees for comparison (*Supplementary files 1 and 2*). Using this approach, more than half of all branches were consistently supported in PB2, PB1, and HA trees; however, NS trees remained largely unsupported regardless of the sequence identity cutoff selected. Branch support varied between replicate trees of PA, NP, and NA, with no single replicate yielding consistently high branch support across vRNA trees. Therefore, we analyzed all seven replicate trees for each of the eight vRNA segments, for a total of 56 trees analyzed from each set of H3N2 viruses (*Figure 1C*, *Figure 1—figure supplement 1*, and *Supplementary files 1 and 2*).

Tree similarity among different vRNA segments is expected to be highest when there are strong epistatic interactions between encoded protein and/or RNA complexes (*Kryazhimskiy et al., 2011*; *Neverov et al., 2015*; *Nshogozabahizi et al., 2017*). We expected to observe such epistasis between protein subunits of the heterotrimeric polymerase complex such as the PB1 and PA segments (*Fodor, 2013*), whereas we did not expect to observe epistasis between PB1 and HA, which do not share any known protein function. Therefore, we examined the extent of similarity between the PB1 tree and the PA and HA trees in H3N2 viruses from 2005 to 2014. Trees built from the PB1 and PA segments had low tree distances (RF = 6 and CID = 0.25) (*Figure 2A*), suggesting that these genes evolve in parallel. PB1 and HA trees from the same set of H3N2 strains had higher tree distances (RF = 14 and CID = 0.44) (*Figure 2B*), suggesting that parallel evolution between PB1 and HA is weaker than that of PB1 and PA. These data are consistent with known protein interactions between PB1 and PA and suggest that tree similarity can be used to identify direct intermolecular interactions that constrain evolution, leading to converging evolutionary trajectories in the trees. Thus, pairwise tree distances recapitulate anticipated protein-driven parallel evolution between two influenza proteins.

Genome-wide inferences of tree similarity can distinguish the relative degree of parallel evolution of all eight genomic segments to each other and capture the strongest overall relationships between segments. To examine the extent of parallel evolution between all vRNA segments, we measured tree distances in all sets of vRNA trees from H3N2 viruses from 2005 to 2014. *Figure 2C* shows the overall mean tree distances between each pair of vRNA segments as determined by RF (refer to *Figure 2—figure supplement 1* for the standard error of the mean, or SEM, between sets of trees).

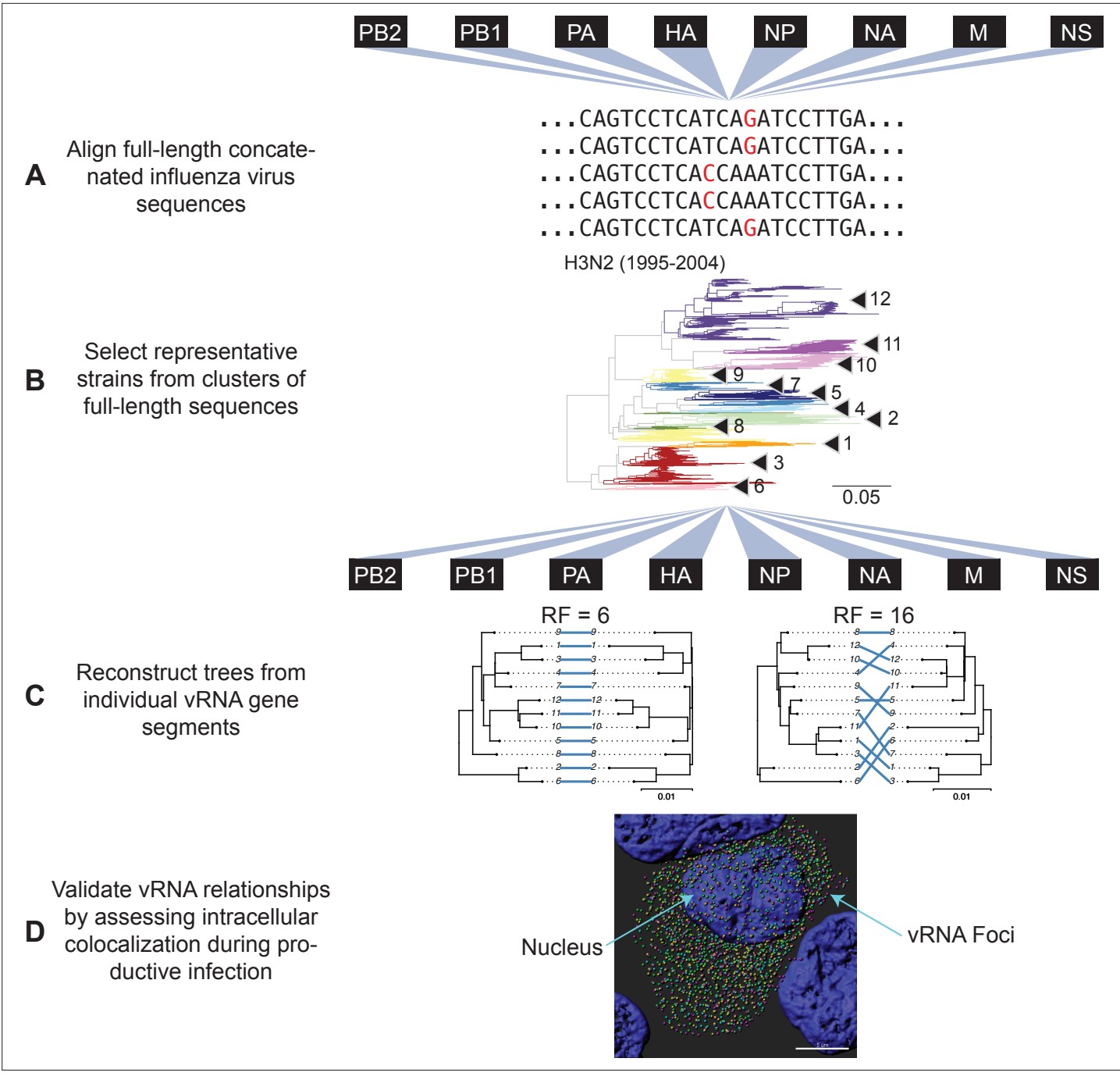

**Figure 1.** Experimental overview. (**A**) Human H3N2 or H1N1 virus sequences were downloaded from the Influenza Research Database and subset into two time periods each: 1995 to 2004 and 2005 to 2014 (H3N2 viruses); 2000 to 2008 and 2010 to 2018 (H1N1 viruses). The H3N2 virus dataset (1995 to 2004) is illustrated here. All eight viral RNA (vRNA) segments from each strain were concatenated into a full-length genome from which alignments were made. (**B**) A species tree was built clustering strains into operational taxonomic unit with at least 97 % sequence identity. Arrowheads denote clusters 1–12. Seven replicate strains were randomly selected from each cluster for further analysis. (**C**) Full-length genomic sequences were partitioned into individual vRNA gene sequence alignments and maximum-likelihood phylogenetic trees were reconstructed from each vRNA gene segment in each replicate. Tree similarity was determined by the Robinson-Foulds distance (RF) and clustering information distance (CID) in each pair of trees. Left, a pair of highly similar trees with a low tree distance plotted in a tanglegram (e.g. back-to-back trees), with intersecting blue lines matching tips. Right, a tanglegram of a pair of dissimilar trees with a high tree distance. Scale bars indicate substitutions per site. (**D**) Colocalization of vRNA segments exhibiting high and low tree similarity were assessed by fluorescence in situ hybridization (FISH). Cells were infected with viruses representative of those analyzed in (**A**–**C**) and fixed and stained with FISH probes specific for vRNA segments of interest. Cells were imaged using confocal microscopy and colocalization between vRNA segments was quantified.

*Figure 1 continued on next page*

*Figure 1 continued*

The online version of this article includes the following figure supplement(s) for figure 1:

**Figure supplement 1.** Genomic viral RNA (vRNA) segment trees.

**Figure supplement 2.** Full-length concatenated genome trees.

To establish a threshold for significance of tree distances, we determined a 95 % confidence interval for RF using a null dataset of randomly generated trees with an equivalent number of leaves (12 in this case, *Figure 2—figure supplement 3A,D*). Low tree distances rarely occurred by chance, with the vast majority of tree distances being greater than 15 in null trees. By comparison, the mean RF of vRNA trees ranged from 6.5 (PB1 and PA) to 15 (PA and NS). Surprisingly, the PB2 tree shared the highest similarity with the NA tree rather than the PB1 or PA trees, suggesting that the relationship between PB2 and NA may supercede the essential role of the PB2 protein in the polymerase complex. In contrast, the mean RF of the NS trees with most other vRNA trees were 14–15, approaching the 95 % confidence threshold of 15.3. However, distances between the NS tree and the other vRNA trees were difficult to interpret, owing to a lack of branch support in NS trees (*Figure 1—figure supplement 1B*). Pairwise intersegmental relationships determined by RF were remarkably reproducible when compared to CID (*Figure 2—figure supplement 2*). To further visualize relationships between all eight vRNA segments, we assembled networks of the pairwise tree distances (*Figure 2—figure supplement 4*). These networks reveal robust parallel evolution between PB1, PA, NP, and NA in H3N2 viruses.

## Evolutionary relationships are largely conserved over time within H3N2 viruses

Recent studies have identified a highly plastic and redundant network of interactions between vRNA segments in influenza virus particles produced during productive infection, many of which may be transient (*Dadonaite et al., 2019*; *Le Sage et al., 2020*). Based on these observations, it is plausible that vRNA relationships identified using our methods might change over time. To examine whether the shared evolutionary trajectories we observed in H3N2 viruses are conserved, we estimated tree distances between all pairs of vRNA trees in H3N2 viruses from an earlier time period (1995–2004) (mean RF: *Figure 3A*; SEM of RF: *Figure 3—figure supplement 1*; mean CID: *Figure 3—figure supplement 2A*; SEM of CID: *Figure 3—figure supplement 2B*). As was seen in H3N2 viruses from 2005 to 2014, tree distances ranged widely, with the highest tree similarity found between PB1, PA, NP, and NA trees of this time period. Networks constructed from pairwise distances that visualize the overall relatedness of vRNA segments confirm that PB1, PA, NA, and NP share the closest distances overall (*Figure 3—figure supplement 3*). Statistical differences between RF from each time period were only found for the NS segment (*Figure 3C*; refer to *Supplementary file 6* for exact p-values). However, NS trees had consistently low bootstrap support (*Figure 1—figure supplement 1A,B*), so these differences may be attributable to insufficient resolution in the underlying trees. Finally, tree distances for H3N2 viruses from 1995 to 2004 were positively correlated with those from 2005 to 2014 (*Figure 3B* and *Figure 3—figure supplement 4*). Taken together, we conclude that phylogenetic relationships between vRNA segments in H3N2 viruses are largely conserved across these two time periods.

## Evolutionary relationships between vRNA segments are dependent upon subtype and lineage

Our results suggest that vRNA relationships are remarkably consistent across H3N2 viruses from a period spanning two decades. To examine whether our approach captures anticipated changes in vRNA relationships in seasonal human influenza viruses of other subtypes and lineages, we assessed these relationships in H1N1 viruses from 2000 to 2008 and 2010 to 2018. Human H1N1 viruses from these time periods represent distinct lineages before and after the 2009 pandemic. This pandemic was caused by an antigenically shifted H1N1 virus that emerged from reassortment of two swine-origin viruses, the North American triple reassortant swine H1N1 virus and Eurasian swine H1N1 virus (*Garten et al., 2009*). Therefore, different evolutionary relationships between vRNA segments would be expected for each lineage.

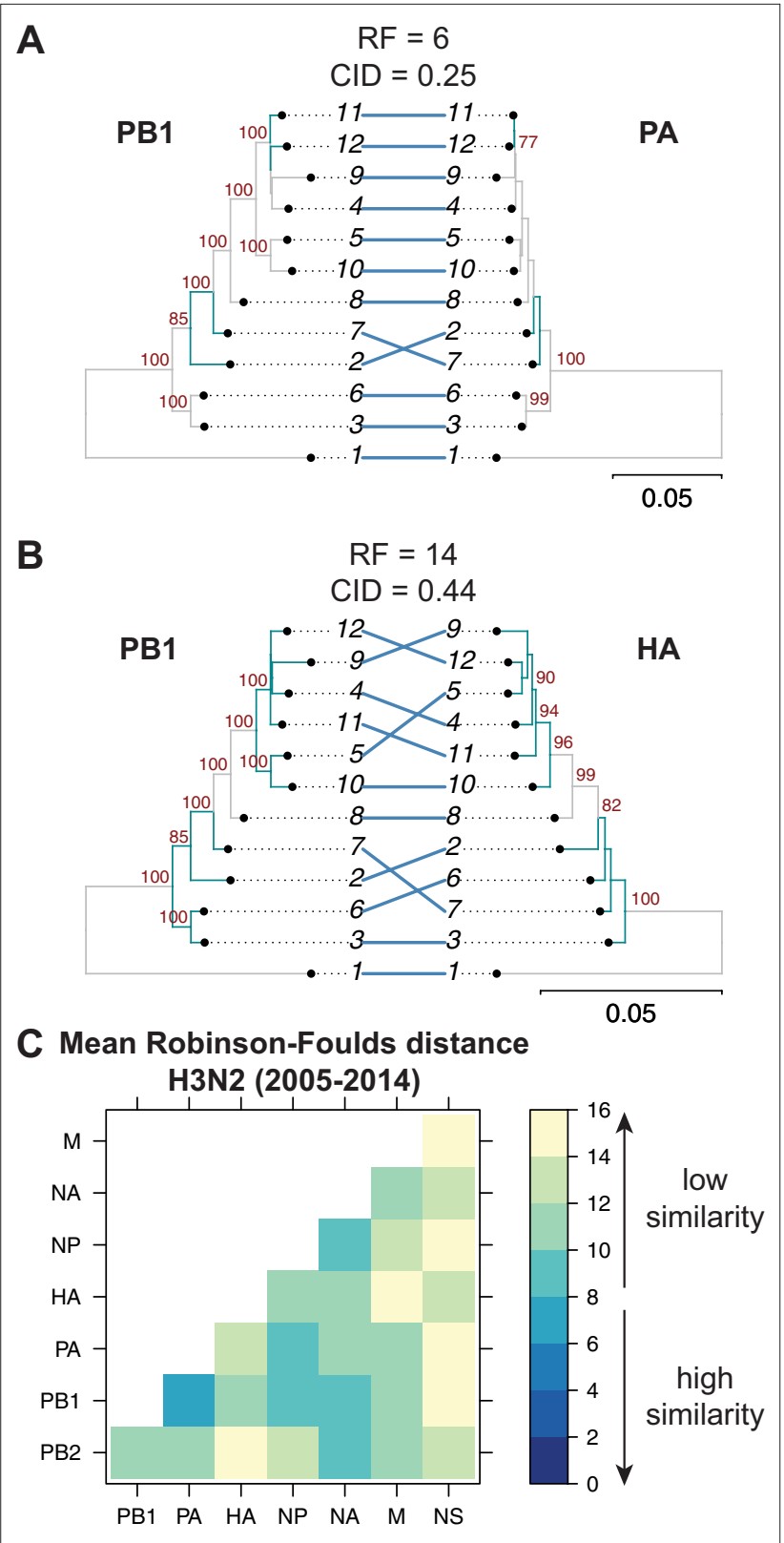

**Figure 2.** Parallel evolution between viral RNA (vRNA) segments varies in H3N2 viruses from 2005 to 2014. Seven replicate maximum-likelihood trees were reconstructed for each vRNA gene segment from human H3N2 virus sequences (2005 to 2014) as described in *Figure 1*. (**A–B**) Highly similar (PB1 and PA gene segments) (**A**) or dissimilar (PB1 and HA gene segments) trees (**B**) from replicate one were plotted as tanglegrams

*Figure 2 continued on next page*

*Figure 2 continued*

with discrepancies in branch topology highlighted in green. Robinson-Foulds distances (RF) and clustering information distances (CID) are shown above the tanglegram. Intersecting lines map leaves on the left tree to the corresponding leaves on the right. Strains are coded by cluster number; strain identities can be found in *Supplementary file 2*. Bootstrap values greater than 70 are shown in red. Scale bars indicate substitutions per site. (**C**) Pairwise RF were calculated between each pair of trees in each replicate. Mean tree distances were visualized in a heatmap. Refer to *Figure 2—figure supplement 1* for the standard error of the mean RF of each pair of trees.

The online version of this article includes the following figure supplement(s) for figure 2:

**Source data 1.** Mean Robinson-Foulds distance (RF) of pairwise replicate tree comparisons of H3N2 viruses from 2005 to 2014 (corresponding to *Figure 2C*).

**Figure supplement 1.** The standard error of the mean (SEM) of replicate Robinson-Foulds distances (RF).

**Figure supplement 2.** The mean clustering information distance (CID) of replicate viral RNA (vRNA) trees.

**Figure supplement 2—source data 1.** Pairwise clustering information distance (CID) for each replicate tree from H3N2 viruses from 2005 to 2014.

**Figure supplement 3.** Null distribution of Robinson-Foulds distances (RF).

**Figure supplement 4.** Networks determined from pairwise tree distances.

Species trees comprising full-length concatenated H1N1 virus genomes from 2000 to 2008 or 2010 to 2018 were constructed and clusters were defined using the same approach described for H3N2 viruses (*Figure 1A–B*). While this method produced a similar number of clusters for both sets of H1N1 viruses (*Table 1*), there were fewer clusters than in H3N2 viruses, owing to the higher rate of evolution observed in H3N2 viruses (*Bedford et al., 2015*). Seven strains were selected from each cluster (*Supplementary files 3 and 4*) and replicate vRNA trees were built as in *Figure 1C*, *Figure 1—figure supplement 1C,D*. *Figures 4A and 5A* show the mean RF for each pair of vRNA segments in H1N1 viruses from 2000 to 2008 and 2010 to 2018, respectively (SEM: *Figure 4—figure supplement 1* and *Figure 5—figure supplement 1*). These heatmaps suggested that tree distances were very different for H1N1 viruses when compared to H3N2 viruses. Using linear regression, we found that tree distances from H1N1 viruses shared either a modest negative correlation or none at all with H3N2 viruses from either time period (*Figure 4B and C*, and *Figure 5—figure supplement 3*). Tree distances determined by CID (2000–2008 mean and SEM: *Figure 4—figure supplement 2A,B*, respectively; 2010–2018 mean and SEM: *Figure 5—figure supplement 2A,B*, respectively) likewise indicated similar trends (*Figure 4—figure supplement 3* and *Figure 5—figure supplement 4A,B*). This is further supported by networks constructed from the pairwise distances for H1N1 viruses as compared to those from H3N2 viruses. The distance networks from H3N2 viruses suggest highest overall tree similarity between PB1, NP, PA, and NA (*Figure 2—figure supplement 4*, *Figure 3—figure supplement 3*). In contrast, the networks from pre-pandemic H1N1 viruses indicate highest tree similarity between PB1, NP, M, and NS (*Figure 4—figure supplement 4*). Networks from post-pandemic H1N1 viruses likewise reflect a different pattern in tree relatedness from that seen in H3N2 virus networks (*Figure 5—figure supplement 5*). Therefore, our data suggest that parallel evolution between vRNA segments overall have significantly diverged between seasonal human influenza H1N1 and H3N2 viruses from similar time scales.

Heatmaps comparing tree distances between vRNA pairs further suggested that vRNA relationships are not conserved across H1N1 viruses of different lineages (*Figure 4A* vs. *Figure 5A*). Linear regression comparing tree distances between vRNA segments from pre-pandemic and post-pandemic H1N1 viruses confirmed no correlation between these trees (*Figure 5B* and *Figure 5—figure supplement 4C*). To examine individual differences between pairs of vRNA trees in H1N1 viruses of different lineages, we plotted RF from pre-pandemic H1N1 viruses alongside RF from post-pandemic H1N1 viruses (*Figure 5C*; refer to *Supplementary file 6* for exact p-values). The 95 % confidence interval cutoff for RF corresponding to trees with nine leaves was 8.6 (*Figure 2—figure supplement 3C*) and is the threshold used for statistical comparison of parallel evolution in vRNA segments from pre-pandemic and post-pandemic H1N1 strains. In stark contrast to the relatively conserved vRNA relationships observed in H3N2 viruses over time, many relationships between vRNA segments were disrupted in post-pandemic H1N1 viruses. Parallel evolution between PB1 and NP observed in pre-pandemic H1N1 viruses (mean RF increased from 1 to 5) was notably displaced by stronger coevolution

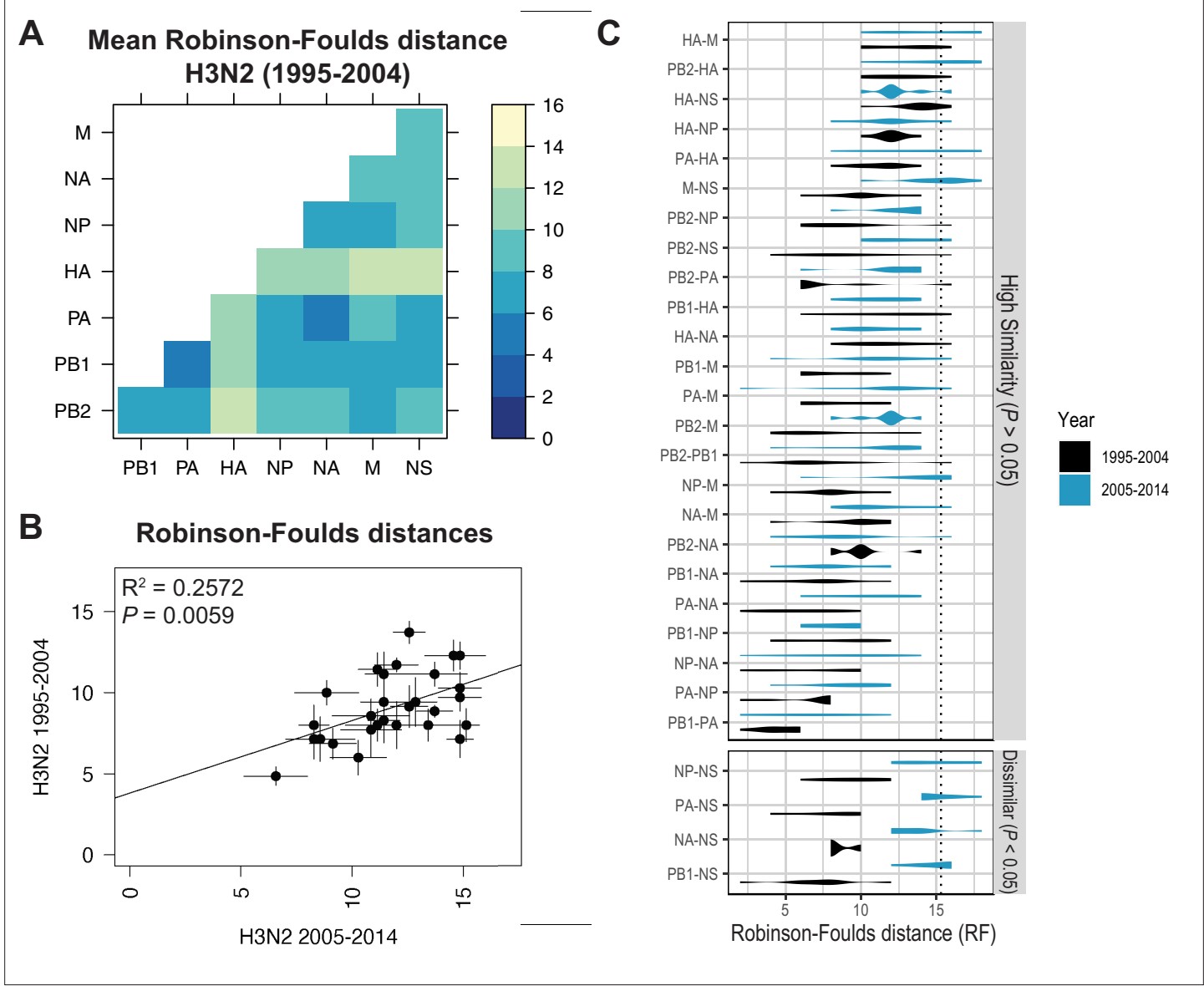

**Figure 3.** Parallel evolution of viral RNA (vRNA) segments from H3N2 viruses is conserved through antigenic drift. (**A**) Seven replicate maximum-likelihood trees were reconstructed for each vRNA gene segment from human H3N2 virus sequences (1995 to 2004) as described in *Figure 1*. Pairwise Robinson-Foulds distances (RF) were calculated between each pair of trees in each replicate. Mean tree distances were visualized in a heatmap. Refer to *Figure 3—figure supplement 1* for the standard error of the mean (SEM) of each pair. (**B**) Mean RF of replicate trees from H3N2 viruses from 1995 to 2004 were plotted against those from 2005 to 2014. The line of best fit was determined by linear regression (solid line). The $R^2$ and p-value are indicated. Error bars indicate the SEM of all replicates. (**C**) Replicate tree distances were plotted comparing H3N2 viruses from 1995 to 2004 (black) to H3N2 viruses from 2005 to 2014 (turquoise). 'Dissimilar' pairs are grouped where $p < 0.05$ (Mann-Whitney *U* test with Benjamini-Hochberg correction; exact p-values reported in *Supplementary file 6*). Dashed line, 95 % confidence interval for tree similarity (determined by a null dataset; refer to *Figure 2—figure supplement 3*).

The online version of this article includes the following figure supplement(s) for figure 3:

**Source data 1.** Mean Robinson-Foulds distance (RF) of pairwise replicate tree comparisons of H3N2 viruses from 1995 to 2004 (corresponding to *Figure 3A*).

**Source data 2.** Pairwise Robinson-Foulds distance (RF) for each replicate tree from H3N2 viruses from 1995 to 2004 or 2005 to 2014, as indicated (corresponding to *Figure 3B and C*).

**Figure supplement 1.** The standard error of the mean (SEM) of replicate Robinson-Foulds distances (RF).

**Figure supplement 2.** The mean clustering information distance (CID) of replicate viral RNA (vRNA) trees.

**Figure supplement 2—source data 1.** Pairwise clustering information distance (CID) for each replicate tree from H3N2 viruses from 1995 to 2004.

*Figure 3 continued on next page*

of PB1 with NA in post-pandemic H1N1 viruses (mean RF decreased from 9 to 3). The M and NS trees shared similar topologies across H1N1 lineages, but each one was significantly more coevolved with the HA and NA trees in post-pandemic viruses. The PB2 trees diverged significantly from the PA trees in favor of greater parallel evolution with the NP and NS trees in post-pandemic H1N1 viruses. Some of these data can be explained by weaker bootstrap support in H1N1 trees, particularly those from H1N1 viruses from 2010 to 2018 (*Figure 1—figure supplement 1C,D*). However, these data imply that shared evolutionary trajectories have significantly diverged between H1N1 lineages. The genomic segments of the swine-origin 2009 pandemic H1N1 virus were contributed by human, avian, and swine hosts (*Garten et al., 2009*). Therefore, our data suggest that host origin may impact the evolutionary trajectory of emerging reassortant viruses from different lineages and the resultant relationships between genomic segments of contemporary H1N1 viruses in humans.

## Parallel evolution in H3N2 viruses is not driven solely by protein-coding mutations

As discussed previously, shared evolutionary trajectories could arise from either protein-protein or RNA-RNA interactions. We have already shown that known protein relationships between PB1 and PA, two members of the polymerase complex, are captured by our approach (*Figure 2A*). However, the observation that PB2, another member of the polymerase complex, is more coevolved with NA than with either PB1 or PA (*Figure 2C*) suggests that our method also reveals protein-independent parallel evolution, since these proteins are not known to function together during infection. Using H3N2 viruses from 2005 to 2014, which yielded vRNA trees with the highest overall bootstrap support (*Figure 1—figure supplement 1B*), we explored the extent to which parallel evolution between vRNA segments is driven by protein-coding mutations. To do so, we converted the vRNA sequence alignments, which are negative-sense, into positive-sense RNA (i.e. coding sense) and translated the coding sequences into amino acid alignments. For the M and NS sequence alignments that encode two splice variants each, the M1/M2 and NS1/NS2 amino acid alignments were both translated. Neighbor-joining trees were reconstructed from the amino acid alignments and the evolutionary relationships between H3N2 proteins were analyzed by RF. We constructed a network from the resultant RF between all pairs of protein trees as was previously done with vRNA trees (*Figure 6—figure supplement 1*). This network appears distinct from networks built from the corresponding gene (vRNA) trees (*Figure 2—figure supplement 4*). As might be expected, the greatest degree of parallel evolution lying at the core of this network was between HA and NA, two viral glycoproteins with coordinated functions in attachment, motility, and entry (*Bloom et al., 2010*; *Sakai et al., 2017*).

To compare parallel evolution between influenza proteins to that of the parent vRNA segments, the mean RF from the gene trees were plotted against the mean RF from the protein trees (*Figure 6*). In the case of the M and NS segments, the mean RF of all protein trees derived from the same gene (i.e. M1/M2 or NS1/NS2) were plotted against the mean RF of the corresponding gene trees. Many vRNA pairs, such as the polymerase subunits PB2 and PB1, lie along the identity line, indicating that protein interactions are more likely to drive parallel evolution in those vRNA segments. Interestingly, HA and NA were the only pair of vRNA segments that lay significantly above the identity line, strongly supporting the observation made by others that epistatic interactions between these proteins constrain their evolution (*Jang and Bae, 2018*; *Kryazhimskiy et al., 2011*; *Neverov et al., 2015*). Of particular interest was that several vRNA segments, such as PB2 and NA (*Figure 6*, open diamond), lay significantly below the identity line. This could be indicative of either purifying selection or of greater parallel evolution between the vRNA segments than the proteins encoded. While this is not altogether unexpected, considering that the mutation rate of a protein is unlikely to be as high as the mutation rate of the corresponding gene, we would expect conserved RNA interactions to also have this effect. These observations suggest that parallel evolution may identify putative RNA interactions between vRNA segments that could facilitate selective assembly and packaging.

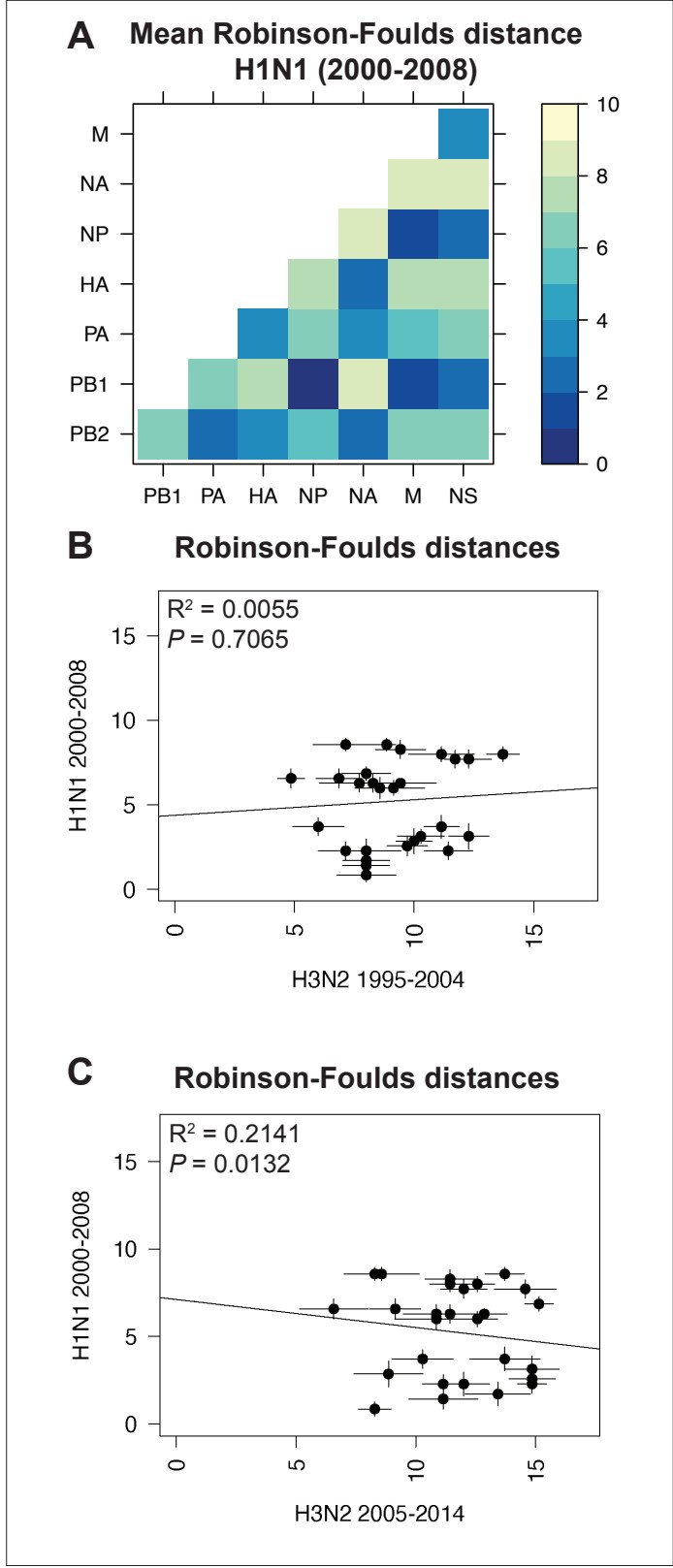

**Figure 4.** Parallel evolution between viral RNA (vRNA) segments is dependent upon subtype. (**A**) Seven replicate maximum-likelihood trees were reconstructed for each vRNA gene segment from human H1N1 virus sequences from 2000 to 2008 as described in *Figure 1*. The pairwise Robinson-Foulds distance (RF) between trees was calculated for each set of replicate trees. Mean distances were visualized in a heatmap. Refer to *Figure 4—figure*

*Figure 4 continued on next page*

*Figure 4 continued*

*supplement 1* for the standard error of the mean (SEM) of each pair. (**B–C**) Mean RF of replicate trees from H1N1 viruses from 2000 to 2008 were plotted against those from H3N2 viruses from 1995 to 2004 (**B**) and H3N2 viruses from 2005 to 2014 (**C**). The line of best fit was determined by linear regression (solid line). The $R^2$ and p-value are indicated. Error bars indicate the SEM of all replicates.

The online version of this article includes the following figure supplement(s) for figure 4:

**Source data 1.** Mean Robinson-Foulds distance (RF) of pairwise replicate tree comparisons of H1N1 viruses from 2000 to 2008 (corresponding to *Figure 4A*).

**Figure supplement 1.** The standard error of the mean (SEM) of replicate Robinson-Foulds distances (RF).

**Figure supplement 2.** The mean clustering information distance (CID) of replicate viral RNA (vRNA) trees.

**Figure supplement 2—source data 1.** Pairwise clustering information distances (CID) for each replicate tree from H1N1 viruses from 2000 to 2008.

**Figure supplement 3.** Linear regression of tree distances determined by clustering information distances (CID).

**Figure supplement 4.** Networks determined from pairwise tree distances.

## PB2 and NA viral ribonucleoprotein complexes preferentially colocalize at the nuclear periphery in vitro

To address whether parallel evolution between the PB2 and NA segments corresponds with their behavior during influenza virus infection, we examined whether these vRNA segments preferentially colocalize in infected cells (*Figure 1D*). During influenza virus infection, vRNA are synthesized in the nucleus, bound by NP in viral ribonucleoprotein (vRNP) complexes, and then transported to the plasma membrane for packaging on endocytic vesicles (*Lakdawala et al., 2016*). Direct RNA-RNA interactions are thought to drive selective assembly of all eight vRNA segments into virus particles, with a hierarchy existing between interactions (*Dadonaite et al., 2019*; *Le Sage et al., 2020*; *Le Sage et al., 2018*; *Lee et al., 2017*; *Marsh et al., 2008*). Previous studies examining the intracellular localization of vRNA segments demonstrated that after genomic replication, vRNA segments are exported from the nucleus as incomplete subcomplexes, or assembly intermediates (*Lakdawala et al., 2014*). The formation of complete complexes containing all eight segments occurs en route to the plasma membrane through dynamic fusion or fission of vRNA segments (*Bhagwat et al., 2020*; *Lakdawala et al., 2014*). Taken together, these data suggest that interactions between some vRNA segments may serve as a scaffold that facilitates formation of complete complexes of all eight vRNA segments. Our network analyses suggest a putative hierarchy that could in part reflect the proposed hierarchical nature of genomic assembly (*Figure 2—figure supplement 4*). We theorized that those vRNA segments that exhibit high gene tree similarity might preferentially form subcomplexes soon after nuclear synthesis. Using our extensive expertise in visualizing intracellular localization of vRNA segments (*Lakdawala et al., 2014*; *Nturibi et al., 2017*), we examined the localization of three vRNA segments (PB2, NA, and NS) in the context of H3N2 virus infection. These segments encompass a pair with high gene-based parallel evolution (PB2-NA) as well as pairs with less evidence of parallel evolution (PB2-NS; NA-NS) (*Figures 2C and 6*, open diamonds).

Quantification of colocalized vRNA segments was performed using fluorescence in situ hybridization (FISH) and immunofluorescence (IF) in productively infected cells (*Lakdawala et al., 2014*; *Nturibi et al., 2017*). Lung epithelial A549 cells were infected for 8 hr with a seasonal human H3N2 virus representative of the time period analyzed (A/Perth/16/2009) and stained for three vRNA segments, NP, and nuclei. The NP antibody stain was used to normalize pairwise colocalization data to the total number of vRNP foci present in cells. Entire cell volumes were captured and the nucleus was masked to analyze colocalization of vRNA segments specifically within the cytoplasm. A representative image of an infected cell from one of three independently performed experiments is shown after processing (*Figure 7A*) and at various stages of image analysis (*Figure 7B*).

Whole cytoplasmic analysis of vRNP colocalization in 15 individually analyzed cells revealed that the majority of cytoplasmic foci contained all three vRNA segments (*Figure 7C*). These data may represent heterogeneity in genomic assembly: whole cytoplasmic analysis is likely to capture vRNP subcomplexes at various stages of assembly, regardless of whether direct RNA-RNA interactions underlie colocalization (*Lakdawala et al., 2014*). In contrast, perinuclear assembly intermediates are more likely to reflect essential RNA-RNA interactions (*Majarian et al., 2018*). Therefore, we assessed

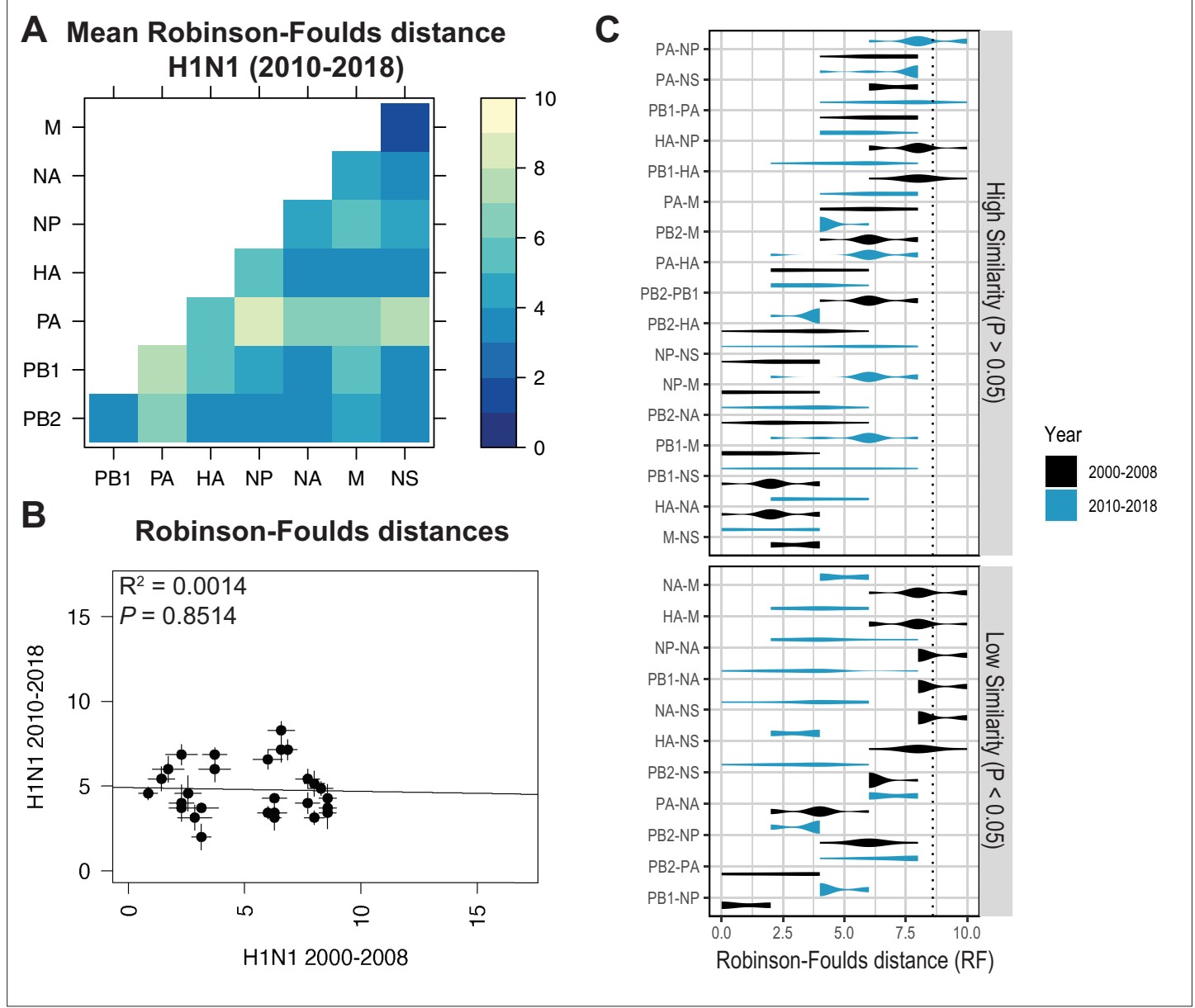

**Figure 5.** Parallel evolution between viral RNA (vRNA) segments diverges in antigenically shifted H1N1 viruses. (**A**) Seven replicate maximum-likelihood trees were reconstructed for each vRNA gene segment from human H1N1 virus sequences from 2010 to 2018 as described in *Figure 1*. The pairwise Robinson-Foulds distance (RF) between trees was calculated for each set of replicate trees. Mean tree distances were visualized in a heatmap. Refer to *Figure 5—figure supplement 1* for the standard error of the mean (SEM) of each pair. (**B**) Mean RF of replicate trees from H1N1 viruses from 2000 to 2008 were plotted against those from 2010 to 2018. The line of best fit was determined by linear regression (solid line). The $R^2$ and p-value are indicated. Error bars indicate the SEM of all replicates. (**C**) Replicate RF were plotted comparing H1N1 viruses from 2000 to 2008 (black) to H1N1 viruses from 2010 to 2018 (turquoise). 'Low similarity' pairs are grouped where p < 0.05 (Mann-Whitney *U* test with Benjamini-Hochberg correction; exact p-values reported in *Supplementary file 6*). Dashed line, 95 % confidence interval for tree similarity (determined by a null dataset; refer to *Figure 2—figure supplement 3*).

The online version of this article includes the following figure supplement(s) for figure 5:

**Source data 1.** Mean Robinson-Foulds distance (RF) of pairwise replicate tree comparisons of H1N1 viruses from 2010 to 2018 (corresponding to *Figure 5A*).

**Source data 2.** Pairwise Robinson-Foulds distance (RF) for each replicate tree from H1N1 viruses from 2000 to 2008 or 2010 to 2018, as indicated (corresponding to *Figure 5B and C*).

**Figure supplement 1.** The standard error of the mean (SEM) of replicate Robinson-Foulds distances (RF).

**Figure supplement 2.** The mean clustering information distance (CID) of replicate viral RNA (vRNA) trees.

*Figure 5 continued on next page*

*Figure 5 continued*

**Figure supplement 2—source data 1.** Pairwise clustering information distance (CID) for each replicate tree from H1N1 viruses from 2010 to 2018.

**Figure supplement 3.** Linear regression of tree distances determined by Robinson-Foulds distance (RF).

**Figure supplement 4.** Linear regression of tree distances determined by clustering information distances (CID).

**Figure supplement 5.** Networks determined from pairwise tree distances.

the potential for PB2, NA, and NS to colocalize at the nuclear periphery, where assembly intermediates first begin to form. We defined localization at the nuclear periphery to within 300 nm, the limit of resolution in this system. Examination of newly exported vRNP complexes within 300 nm of the nuclear periphery revealed an enrichment of PB2-NA vRNP complexes over either NA-NS or PB2-NS vRNP complexes (*Figure 7D*). These data indicate that PB2 and NA preferentially colocalize with each other after nuclear synthesis and support the hypothesis that parallel evolution between segments can reveal putative RNA-RNA interactions. Moreover, these data implicate RNA-RNA interactions,

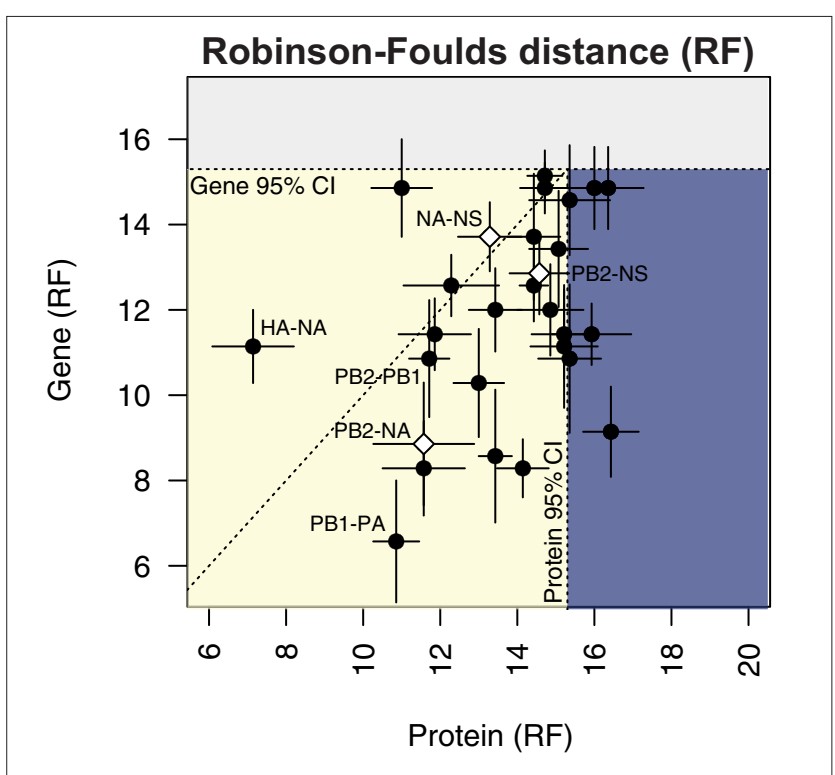

**Figure 6.** Protein-coding substitutions do not fully account for parallel evolution between genes. H3N2 virus viral RNA (vRNA) gene sequence alignments from 2005 to 2014 were translated into the corresponding amino acid alignments. Neighbor-joining trees were reconstructed from these alignments and the Robinson-Foulds distance (RF) was tabulated for all protein tree pairs. The mean tree distance of each pair of protein trees was plotted against the mean tree distance of the corresponding gene trees. For the M and NS gene segments, which encode multiple protein products, tree distances were calculated for each protein tree individually and the average distances are shown. Error bars indicate the standard error of the mean (SEM) of replicate trees. Dashed horizontal and vertical lines, 95 % confidence interval (CI) for tree similarity, as determined by a null dataset (refer to *Figure 2—figure supplement 3*). The region shaded yellow lies within the 95% CI for both gene and protein trees with the identity line plotted. The region shaded blue lies within the 95% CI for gene trees but not protein trees. The region shaded gray lies outside the 95% CI for both gene and protein trees.

The online version of this article includes the following figure supplement(s) for figure 6:

**Source data 1.** Mean Robinson-Foulds distance (RF) of pairwise replicate gene or protein tree comparisons from H3N2 viruses from 2005 to 2014.

**Figure supplement 1.** Parallel evolution between proteins in H3N2 viruses from 2005 to 2014.

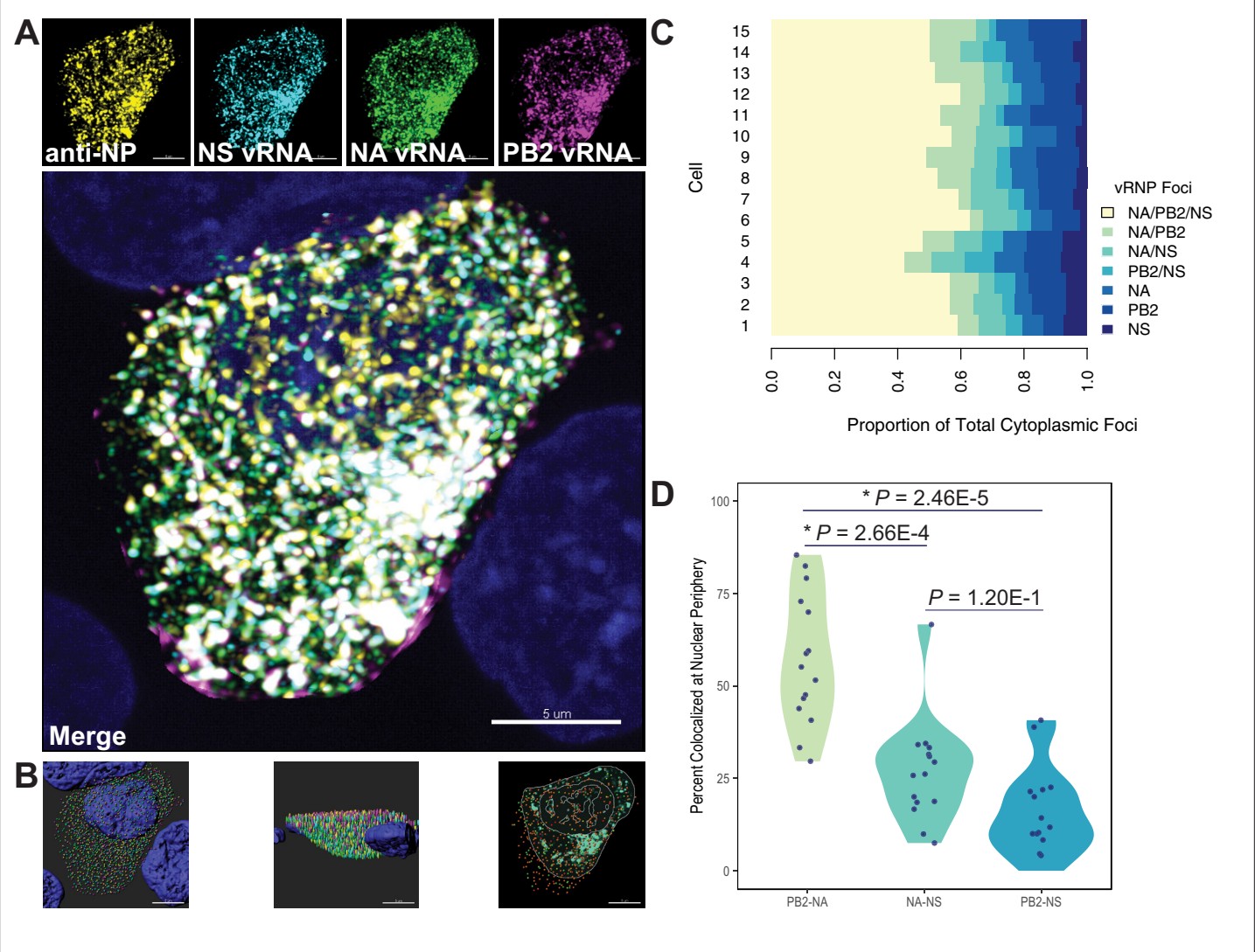

**Figure 7.** Colocalization of viral RNA (vRNA) segments at the nuclear periphery correlates with evolutionary relationships during productive viral infection. A549 cells were infected with A/Perth/16/2009 (H3N2) at a multiplicity of infection (MOI) of 2 or mock-infected. Cells were fixed at 8 hr post-infection and combination fluorescence in situ hybridization/immunofluorescence (FISH and IF, respectively) was performed. FISH probes targeting the NS, NA, and PB2 vRNA segments were labeled with Alexa Fluor 488, Quasar 570, and Quasar 670, respectively. Antibodies targeting nucleoprotein (NP) were used with an anti-mouse Alexa Fluor 594 secondary antibody. Nuclei were labeled with DAPI. Coverslips were mounted and volumetric imaging was performed to obtain Nyquist sampling. (**A**) A maximum projection image of a representative cell is shown after cell segmentation. Scale bar corresponds to 5 μm. (**B**) A 3D rendering of the cell after analysis. (**C**) Colocalization of vRNA segments was assessed in 15 individual infected cells. (**D**) Quantification of each pair of vRNA segments within 300 nm of the nuclear border. Each point represents an individual cell (n = 15). Aggregate data from three independently performed experiments are shown. Asterisks (*) indicate p-adj < 0.05 (Mann-Whitney *U* test with Benjamini-Hochberg correction).

The online version of this article includes the following figure supplement(s) for figure 7:

**Source data 1.** Percent colocalization nucleoprotein (NP)-positive viral RNA (vRNA) foci (NA, PB2, or NS) during productive infection of A549 cells with A/Perth/16/2009 (H3N2).

in addition to protein interactions, as novel drivers of parallel evolution between vRNA segments in seasonal human influenza viruses.

## Discussion

In this study, we used phylogenetics and molecular biology methods to investigate genome-wide relationships between vRNA segments in seasonal human IAV. We found that parallel evolution varies

considerably between vRNA segments, with distinct relationships forming in different influenza virus subtypes (H1N1 vs. H3N2) and between H1N1 virus lineages arising from distinct evolutionary paths. We further demonstrate that evolutionary relatedness between vRNA segments in H3N2 viruses is largely conserved over time. Importantly, our data suggest that parallel evolution cannot be attributed solely to protein interactions, and we successfully predicted intracellular colocalization between two coevolved vRNA segments during infection with an H3N2 virus. Thus, we present a phylogenetic approach for interrogating putative RNA associations that could be broadly applied toward the study of genomic assembly and reassortment in segmented viruses.

Selective assembly of all eight genomic segments is fundamental to the production of fully infectious virus particles. We and others have used a variety of biochemical approaches to investigate the mechanisms that promote selective assembly (*Dadonaite et al., 2019*; *Le Sage et al., 2020*). We previously demonstrated that binding of vRNA segments by NP is non-uniform and non-random (*Le Sage et al., 2018*; *Lee et al., 2017*), supporting the model that intersegmental RNA interactions facilitate selective assembly. Biochemical approaches to define bona fide intersegmental RNA-RNA interactions demonstrated that the interaction network is highly flexible and varies between H1N1 and H3N2 viruses (*Dadonaite et al., 2019*; *Le Sage et al., 2020*). These observations are consistent with our conclusion that RNA interactions constrain parallel evolution between vRNA segments in a manner sensitive to the genetic context studied.

The approach we present here differs from other experimental approaches in that we identify a novel, conserved RNA-driven relationship between vRNA segments in H3N2 viruses. For example, we found that relationships between PB1, PA, NP, and NA are enriched over other segments in H3N2 viruses and conserved over time. One might expect PB1, PA, and NP to coevolve because of the functions of the proteins they encode: the polymerase subunits PB2, PB1, and PA form a supramolecular complex around each vRNA segment with NP (*Fodor, 2013*). However, this explanation does not account for the parallel evolution observed between vRNP components and NA, and our microscopy data demonstrates that the NA segment preferentially colocalizes with the vRNA of one such vRNP component, supporting the possibility that parallel evolution of NA with PB1, PA, and NP could also be driven by RNA-RNA interactions. These observations suggest that RNA relationships with the NA segment may facilitate selective assembly of vRNA segments. Further work should be directed at determining the underlying nature driving the novel relationship between these segments and whether similar assembly intermediates can be identified in H1N1 viruses.

Previous pandemic influenza viruses emerged through reassortment (*Neumann et al., 2009*). Risk assessment for future influenza pandemics relies on understanding assembly of vRNA segments within a cell. As we have discussed, experimental investigations of intersegmental RNA interactions indicate that the vRNA interactome is distinct among virus strains and highly plastic (*Dadonaite et al., 2019*; *Le Sage et al., 2020*). Therefore, experimental approaches are unlikely to provide the holistic view necessary to assess reassortment outcomes of two circulating influenza strains. In contrast, we identified several conserved relationships between vRNA segments in H3N2 viruses that could impose constraints on reassortment. In addition, we identified several key differences between the evolutionary trajectories of vRNA segments in pre-pandemic and post-pandemic H1N1 viruses of different lineages. Experimental investigation of the differences we present here may reveal key vRNA relationships that dictate reassortment and pandemic potential of influenza viruses. Thus, investigation of epistatic relationships between vRNA segments through phylogenetics could inform sequence-based implementation of barriers to reassortment in emerging influenza viruses.

# Materials and methods

**Key resources table**

| Reagent type (species) or resource | Designation | Source or reference | Identifiers | Additional information |
|---|---|---|---|---|
| Gene (influenza A virus) | Seasonal human influenza A virus sequences | Influenza Research Database | Accession numbers provided in *Supplementary files 1–4* | See Materials and methods, Data mining and subsampling section |
| Antibody | Anti-NP (mouse monoclonal) | *Continued on next page* Millipore | Cat# MAB8251, RRID:AB_95293 | IF (1:2000) |

| Reagent type (species) or resource | Designation | Source or reference | Identifiers | Additional information |
|---|---|---|---|---|
| | | *Continued* | | |
| Antibody | Anti-Mouse IgG Alexa Fluor 594 (goat polyclonal) | Invitrogen | Cat# A-11005, RRID:AB_2534073 | IF (1:2000) |
| Recombinant DNA reagent | A/Perth/16/2009 (H3N2) reverse genetics plasmids | PMID:33919124 | | Bidirectional pHW2000 backbone |
| Peptide, recombinant protein | Trypsin, TPCK-treated | Worthington Biochemical | Cat# LS003750 | 1:1000 |
| Cell line (*Homo sapiens*) | A549 cells | ATCC | Cat# CCL-185, RRID:CVCL_0023 | Validation performed by U. of Arizona Genetics Core |
| Sequence-based reagent | H3N2 PB2 FISH probes conjugated to Quasar 670 | This paper | FISH probes | Oligo sequences provided in *Supplementary file 6* |
| Sequence-based reagent | H3N2 NA FISH probes conjugated to Quasar 570 | This paper | FISH probes | Oligo sequences provided in *Supplementary file 6* |
| Sequence-based reagent | Amine-labeled H3N2 NS FISH probes | This paper | FISH probes | Oligo sequences provided in *Supplementary file 6* |
| Commercial assay or kit | Alexa Fluor 488 Oligonucleotide Amine Labeling Kit | Invitrogen | Cat# A20191 | |
| Other | DAPI | Sigma | Cat# D9542 | 0.2 μg/ml |
| Chemical compound, drug | ProLong Diamond antifade mountant | Thermo Fisher | Cat# P36965 | |
| Software, algorithm | R | CRAN | RRID:SCR_001905 | |
| Software, algorithm | Parallel Evolution Of Influenza Viral RNA (custom script) | This paper | | See Materials and methods, Code availability section |
| Software, algorithm | Huygens | Scientific Volume Imaging B.V. | RRID:SCR_014237 | |
| Software, algorithm | Imaris | Bitplane | RRID:SCR_007370 | |
| Software, algorithm | Matlab | MathWorks | RRID:SCR_001622 | |
| Software, algorithm | Matlab extension | PMID:28724771 | | See Materials and methods, Validation of vRNA relationships section |

## Data mining and subsampling

FASTA files of each genomic segment of human IAV sequences of H1N1 and H3N2 viruses were downloaded from the IRD (http://www.fludb.org) (*Zhang et al., 2017*) on June 22, 2018, and July 3, 2018, respectively. Strains lacking full-length genomic sequence data were excluded.

Sequences were read into R (version 3.5.2) using the DECIPHER (version 2.18.1) package (*Wright, 2015*) and subset into the time periods 1995–2004 and 2005–2014 (H3N2 strains) or 2000–2008 and 2010–2018 (H1N1 strains). Time periods were selected in part to ensure a similar level of genetic diversity between strains. In each strain, all eight vRNA segments were concatenated into a full-length genome from which alignments were constructed (*Figure 1A*). A neighbor-joining species tree was built by clustering strains into operational taxonomic units with sequence identity cutoffs ranging from 95% to 99% (*Figure 1B*). In H3N2 viruses from 1995 to 2004, there were 3, 7, 16, 53, and 259 clusters corresponding to cutoffs of 95%, 96%, 97%, 98%, and 99 % sequence identity, respectively. The 95–96% sequence identity cutoffs were discarded, as these produced trees with an insufficient number of branches for comparison by RF. However, as the cutoff for sequence identity was increased

from 97% to 99%, we observed a corresponding decrease in bootstrap support for trees built from representative sequences. A sequence identity cutoff of 97 % was therefore selected to ensure the greatest degree of robustness in tree topologies. Small clusters occurred infrequently and were either omitted or collapsed into a single cluster. Seven replicate strains were randomly chosen from each cluster for further study and visually inspected for sequencing ambiguities. A list of all strains analyzed and the corresponding accession numbers can be found in *Supplementary files 1-4*.

## Analysis of tree similarity

Maximum-likelihood trees were reconstructed under the *Hasegawa et al., 1985* model for either full-length genomes or individual vRNA segments with 100 or 1000 bootstrap replicates, as indicated, using the DECIPHER package in R (*Figure 1C*). The phangorn package (version 2.5.5) (*Schliep, 2011*) was used to identify an appropriate model of evolution for phylogenetic reconstruction. Strain names are coded by cluster number in all trees. Phylogenetic trees of full-length concatenated genomes are shown in *Figure 1—figure supplement 2*. Neighbor-joining protein trees were built from amino acid alignments after translation of the corresponding coding sequence alignments. Networks visualizing overall vRNA relatedness with mean tree distances between vRNA segments were built using the UPGMA method.

Tanglegrams, or back-to-back trees with intersecting lines matching tips from the left tree to the right tree, were built from pairs of vRNA phylogenies within replicates using the phytools package (version 0.7–70) (*Revell, 2012*). RF was calculated for each pair of trees using the ape package (version 5.4–1) (*Paradis and Schliep, 2019*). CID was calculated with the TreeDist package (version 2.0.3) (*Smith, 2020*).

Linear regression was used to determine the overall association between tree distances from different sets of viruses (RF or CID). A set of 1000 randomly sampled, unrooted trees with 8, 9, or 12 tips were built using the ape package to determine confidence intervals for the RF between phylogenetic trees. RF was calculated for all pairs of trees and these were fit to a linear regression model. For visualization purposes, null RF values were either log-transformed or transformed by the Yeo-Johnson method (*Yeo and Johnson, 2000*), as indicated. Mean RF calculated for pairs of vRNA trees were considered significant if they fell within the first five percentiles as compared to null RF from random trees with the same number of tips (i.e. 95 % of null RF were greater than the mean RF for a given pair of vRNA trees). A Mann-Whitney *U* test with a Benjamini-Hochberg post hoc correction was used to identify statistically significant differences between RF from two time periods.

## Validation of vRNA relationships

Human adenocarcinoma alveolar basal epithelial cells (A549, ATCC) were maintained in high-glucose Dulbecco's modified Eagle medium (Sigma) supplemented with 10 % fetal bovine serum (HyClone), 2 % L-glutamine, and 1 % penicillin/streptomycin. Recombinant virus was rescued as previously described (*Lakdawala et al., 2011*). Virus titers were determined by 50 % tissue culture infectious dose (TCID$_{50}$) using the endpoint titration method (*Reed and Muench, 1938*). Validation of cell lines was performed on a routine basis for mycoplasma contamination and cell line purity and identity. Mycoplasma screening was performed by the vendor and annually thereafter. Cell lines tested negative for mycoplasma at time of purchase (Hoechst stain, agar culture, and PCR-based assays) and mycoplasma status is confirmed negative annually using the MycoAlert Mycoplasma Detection Kit (Lonza). The identity and purity of cell lines were verified at the time of purchase (ATCC) and annually thereafter by short tandem repeat profiling (University of Arizona Genetics Core). Documentation of the A549 cell line purity and identity in these studies is available upon request. Low-passage stocks were maintained for no more than 20 passages after thawing to ensure maintenance of a pure cell population.

Custom Stellaris RNA FISH oligonucleotide probes specific for the H3N2 virus NS, NA, and PB2 vRNA segments were purchased from BioSearch Technologies (refer to *Supplementary file 5* for FISH probe sequences). Each custom probe mix comprises 2040 20-mers that span the length of the vRNA segment of interest. Probes with high complementarity against other vRNA segments or positive-sense RNA were excluded during the design process. The NS probe was purchased with a terminal amine group and manually conjugated to the Alexa Fluor 488 fluorophore using the Alexa Fluor 488

Oligonucleotide Amine Labeling Kit (Invitrogen). The NA and PB2 probes were labeled by the manufacturer with the Quasar 570 and Quasar 670 fluorophores, respectively.

Three independent FISH-IF experiments were performed (*Figure 1D*). A549 cells were seeded directly onto 1.5 mm circular coverslips (Fisher Scientific) in tissue culture dishes. The next day, cells were infected at a multiplicity of infection of 2 with A/Perth/16/2009 (H3N2) or mock-infected in diluent. Cells were fixed at 8 hr post-infection with 4 % paraformaldehyde and permeabilized overnight in ice cold 70 % ethanol. Prior to hybridization, cells were rehydrated in wash buffer (10 % formamide and 2 × saline sodium citrate [SSC] in DEPC-treated $H_2O$) and then incubated at 28 °C overnight in hybridization buffer (10 % dextran sulfate, 2 mM vanadyl-ribonucleoside complex, 0.02 % RNA-free BSA, 1 mg/ml *Escherichia coli* tRNA, 2 × SSC, and 10 % formamide in DEPC-treated $H_2O$) with anti-IAV NP antibody (Millipore, 1:2000) and FISH probes. After hybridization, cells were washed and incubated with Alexa Fluor 594 goat anti-mouse (Invitrogen, 1:2000) and DAPI (Sigma, 1:5000) in wash buffer. Coverslips were mounted on slides in ProLong Diamond antifade mountant (Thermo Fisher).

Microscope slides were imaged on a Leica SP8 confocal microscope equipped with a pulsed white light laser as an excitation source and an acousto-optical beam splitter and Leica Hybrid Detectors. All imaging was performed with a 100 × oil immersion objective with a numerical aperture of 1.4. Sequential scanning with a line averaging of 3 between frames was used. To obtain Nyquist sampling, z-stacks of each cell were taken with a step size of 170 nm to achieve a pixel size of 45 nm × 45 nm × 170 nm. The following custom parameters were established using single-color infected controls for sensitive detection of all five fluorophores: 405 nm excitation wavelength ($\lambda_{ex}$) with 0.5 % laser power and a detection range of 415–470 nm (PMT1; DAPI), 488 nm $\lambda_{ex}$ with 10 % laser power and a detection range of 493–540 nm with time gating of 1–6 nanoseconds (ns) (HyD4; Alexa Fluor 488), 582 $\lambda_{ex}$ with 15 % laser power and a detection range of 590–635 nm with time gating of 1.5–6 ns (HyD4; Cal Fluor Red 590), 545 nm $\lambda_{ex}$ with 5 % laser power and a detection range of 545–568 nm with time gating of 1.5–6 ns (HyD4; Quasar 570), 647 nm $\lambda_{ex}$ with 5 % laser power and a detection range of 670–730 nm with time gating of 1.5–6 ns (HyD5; Quasar 670). In each experiment, five volumetric z-stacks were imaged of infected cells and one z-stack was imaged of mock-infected cells.

Background subtraction and deconvolution of confocal images were performed manually for each channel using Huygens Essential software (version 19.04, Scientific Volume Imaging B.V.). In each experiment, images taken of mock-infected cells were deconvolved using the same parameters as those of infected cells. 3D reconstruction and colocalization analysis of the resulting images were performed using Imaris software (version 8.4.2, Bitplane AG) as previously described (*Lakdawala et al., 2014*; *Nturibi et al., 2017*). Briefly, the cell of interest in each image was segmented using the 'Surfaces' and 'Cell' tools in Imaris software. DAPI signal was used to mask nuclear signal from the remaining channels. The 'Spots' tool was then used to populate the reconstructed cell with four different sets of Spots corresponding to foci from each of the remaining channels. In each experiment, the mock infected cell was analyzed in an identical manner and the fluorescence intensity for each channel of the mock-infected cell was used to establish fluorescence intensity thresholds at which 97 % or more of the background signal was removed prior to Spot generation. A modified Matlab extension was then used to quantify spot colocalization using a distance threshold of 300 nm as previously described (*Nturibi et al., 2017*). Colocalization data was imported into the Cell and all data was exported and analyzed in R. A Mann-Whitney *U* test was used to determine statistical significance of FISH-IF colocalization data.

## Code availability

Custom code for analysis of parallel evolution in concatenated, full-length genomic influenza virus sequences is available on GitHub (https://github.com/Lakdawala-Lab/Parallel-Evolution-Of-Influenza-Viral-RNA/ (copy *Jones, 2020* archived at swh:1:rev:27dc83b8eec1f461bbf9ef3f1dbeba61f0514fb3)).

## Acknowledgements

All confocal microscopy imaging was performed at the Center for Biologic Imaging at the University of Pittsburgh. JEJ is supported by a T32 (T32 AI049820) and the Catalyst Award (University of Pittsburgh Center for Evolutionary Biology and Medicine). This work is funded by the National Institutes of Health NIAID (R01 AI139063). We thank members of the Lakdawala and Wright labs for technical support and constructive feedback on this manuscript.

## Additional information

### Funding

| Funder | Grant reference number | Author |
|---|---|---|
| National Institute of Allergy and Infectious Diseases | T32 AI049820 | Jennifer E Jones |
| Center for Evolutionary Biology and Medicine, University of Pittsburgh | | Jennifer E Jones |
| National Institute of Allergy and Infectious Diseases | R01 AI139063 | Seema S Lakdawala |

The funders had no role in study design, data collection and interpretation, or the decision to submit the work for publication.

### Author contributions

Jennifer E Jones, Conceptualization, Data curation, Formal analysis, Funding acquisition, Investigation, Methodology, Software, Supervision, Validation, Visualization, Writing - original draft, Writing – review and editing; Valerie Le Sage, Investigation, Validation, Writing – review and editing; Gabriella H Padovani, Data curation, Formal analysis, Writing – review and editing; Michael Calderon, Data curation, Methodology, Validation, Visualization, Writing – review and editing; Erik S Wright, Conceptualization, Formal analysis, Funding acquisition, Investigation, Methodology, Project administration, Resources, Software, Supervision, Writing – review and editing; Seema S Lakdawala, Conceptualization, Funding acquisition, Methodology, Project administration, Resources, Software, Supervision, Writing – review and editing

### Author ORCIDs

Jennifer E Jones http://orcid.org/0000-0002-9970-1063
Erik S Wright http://orcid.org/0000-0002-1457-4019
Seema S Lakdawala http://orcid.org/0000-0002-7679-2150

### Decision letter and Author response

Decision letter https://doi.org/10.7554/eLife.66525.sa1
Author response https://doi.org/10.7554/eLife.66525.sa2

---

## Additional files

### Supplementary files

• Supplementary file 1. Human H3N2 sequences analyzed from 1995 to 2004 and the corresponding GenBank accession numbers. Human H3N2 sequences from 1995 to 2004 were downloaded from the Influenza Research Database and full-length genomes were concatenated and grouped into operational taxonomic units (numbered 1–12 under Cluster ID) with at least 97 % sequence identity. Representative sequences were selected from these clusters for further analysis. Each vertical column indicates one replicate (seven replicates total).

• Supplementary file 2. Human H3N2 sequences analyzed from 2005 to 2014 and the corresponding GenBank accession numbers. Human H3N2 sequences from 2005 to 2014 were downloaded from the Influenza Research Database and full-length genomes were concatenated and grouped into operational taxonomic units (numbered 1–12 and labeled Cluster ID) with at least 97 % sequence identity. Representative sequences were selected from these clusters for further analysis. Each vertical column indicates one replicate (seven replicates total). Asterisks (*) denote H3N2v (variant) strains.

• Supplementary file 3. Human H1N1 sequences analyzed from 2000 to 2008 and the corresponding GenBank accession numbers. Human H1N1 sequences from 2000 to 2008 were downloaded from the Influenza Research Database and full-length genomes were concatenated and grouped into operational taxonomic units (numbered 1–9 under Cluster ID) with at least 97 % sequence identity. Representative sequences were selected from these clusters for further analysis. Each vertical

column indicates one replicate (seven replicates total).

• Supplementary file 4. Human H1N1 strains analyzed from 2010 to 2018 and the corresponding GenBank accession numbers. Human H1N1 sequences from 2010 to 2018 were downloaded from the Influenza Research Database and full-length genomes were concatenated and grouped into operational taxonomic units (numbered 1–9 under Cluster ID) with at least 97 % sequence identity. Representative sequences were selected from these clusters for further analysis. Each vertical column indicates one replicate (seven replicates total).

• Supplementary file 5. Fluorescence in situ hybridization (FISH) probe sequences. Custom oligonucleotide probes targeting PB2 (A), NA (B), and NS (C) vRNA were designed from A/Panama/2007/1999 (H3N2) virus sequences using the Stellaris probe designer (BioSearch Technologies). Oligos exhibiting significant complementarity against other vRNA segments and/or positive-strand complementarity were excluded.

• Supplementary file 6. p-Values associated with *Figures 3C and 5C*. A Mann-Whitney *U* test was performed to determine whether pairwise Robinson-Foulds distance (RF) of H3N2 viruses from 1995 to 2004 were significantly different from pairwise RF of H3N2 viruses from 2005 to 2014 (left) and whether pairwise RF of H1N1 viruses from 2000 to 2008 were significantly different from pairwise RF of H1N1 viruses from 2010 to 2018 (right). The 'p-adj' indicates adjusted p-values after Benjamini-Hochberg correction.

• Transparent reporting form

### Data availability

All data generated or analysed during this study are included in the manuscript and supporting files. Scripts are availble on GitHub (https://github.com/Lakdawala-Lab/Parallel-Evolution-Of-Influenza-Viral-RNA/ copy archived at https://archive.softwareheritage.org/swh:1:rev:27dc83b8eec1f461bbf9ef3f1dbeba61f0514fb3). Summary tables are provided for Figures 2-7 and figure supplements.

The following previously published datasets were used:

| Author(s) | Year | Dataset title | Dataset URL | Database and Identifier |
|---|---|---|---|---|
| Aevermann BD, Anderson TK, Burke DF, Dauphin G, Gu Z, He S, Kumar S, Larsen CN, Lee AJ, Li X, Macken C, Mahaffey C, Pickett BE, Reardon B, Smith T, Stewart L, Suloway C, Sun G, Tong L, Vincent AL, Walters B, Zaremba S, Zhao H, Zhou L, Zmasek C, Klem EB, Scheuermann RH, Zhang Y | 2017 | Influenza Research Database: An integrated bioinformatics resource for influenza virus research | http://www.fludb.org/ | Influenza Research Database, 10.1093/nar/gkw857 |

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
