## [Decision Letter]

**Acceptance summary:**

This interesting study provides phylogenetic and molecular evidence for novel RNA-RNA interactions driving the genomic coevolution of Influenza virus subtypes. The authors make important contributions towards understanding how certain genetic combinations can lead to emerging pathogen variants with the possibility of new antigenic properties and spillover.

**Decision letter after peer review:**

Thank you for submitting your article "Parallel evolution between genomic segments of seasonal human influenza viruses reveals RNA-RNA relationships" for consideration by *eLife*. Your article has been reviewed by 3 peer reviewers, one of whom is a member of our Board of Reviewing Editors, and the evaluation has been overseen by George Perry as the Senior Editor. The following individual involved in review of your submission has agreed to reveal their identity: Debapriyo Chakraborty (Reviewer #2).

Essential revisions:

In this study, the authors combined phylogenetics and molecular assays to demonstrate parallel evolution across vRNA segments in influenza A virus subtypes, highlighting the importance of RNA-RNA interaction as a novel driver of viral genomic evolution. While reviewers agree that the work is important and likely to have implications for pathogen evolution and emerging infections, there are several major concerns that have been raised in terms of clarity of working hypotheses, methods and interpretations.

To cite a few, for example, the rationale behind (1) leaving out sequences that did not resolve well in the phylogenic analyses; (2) relating RNA segment co-localization to similar phylogenetic patterns was unclear and thus, requires more qualification.

Unless all the queries raised by the reviewers are addressed, at present, it is difficult to truly judge on its content and what is supported by the data produced.

*Reviewer #1 (Recommendations for the authors ):*

In this paper, authors did a fine job of combining phylogenetics and molecular methods to demonstrate the parallel evolution across vRNA segments in two seasonal influenza A virus subtypes. They first estimated phylogenetic relationships between vRNA segments using Robinson-Foulds distance and identified the possibility of parallel evolution of RNA-RNA interactions driving the genomic assembly. This is indeed an interesting mechanism in addition to the traditional role for proteins for the same. Subsequently, they used molecular biology to validate such RNA-RNA driven interaction by demonstrating co-localization of vRNA segments in infected cells. They also showed that the parallel evolution between vRNA segments might vary across subtypes and virus lineages isolated from distinct host origins. Overall, I find this to be excellent work with major implications for genome evolution of infectious viruses; emergence of new strains with altered genome combination.

I am wondering if leaving out sequences (not resolving well) in the phylogenic analysis interferes with the true picture of the proposed associations. What if they reflect the evolutionary intermediates, with important implications for the pathogen evolution which is lost in the analyses?

Lines 50-51: Can you please elaborate? I think this might be useful for the reader to better understand the context. Also, a brief description on functional association between different known fragments might instigate curiosity among the readers from the very beginning. At present, it largely caters to people already familiar with the biology of influenza virus.

Lines 95-96 Were these strains all swine-origin? More details on these lineages will be useful for the readers.

Lines 128-132: I think it will be nice to talk about these hypotheses well in advance, may be in the Introduction, with more functional details of viral segments.

Lines 134-136: Please rephrase this sentence to make it more direct and explain the why. E.g. "… parallel evolution between PB1 and HA is likely to be weaker than that of PB1 and PA".

Lines 222-223: Please include a set of hypotheses to explain you results? Please add a perspective in the discussion on how this contribute might to the pandemic potential of H1N!?.

Lines 287-288: I am wondering how likely is this to be true for H1N1.

*Reviewer #2 (Recommendations for the authors):*

Abstract: Please define parallel evolution and mention the methods used, without details, at the beginning.

Line 29: "Genetic variation is ubiquitious in RNA viruses." I find this sentence vague. I would prefer a clear statement about high genetic variation in RNA viruses.

Line 38: Typo, concomitant

Line 38: "Thus, the emergence of pandemic strains is marked by a concommittent alteration in the influenza virus genome triggered by new genetic diversity." – This is a very complex sentence; either simplify or break it into two, please.

Line 41: "Genetic mutation is driven by stochastic processes and is therefore difficult to predict (Andino & Domingo, 2015)." – The relevance is not clear. Either remove or provide more context.

Line 50-52: "It is consequently imperative to identify the evolutionary constraints imposed by intersegmental vRNA interactions, as this may help predict future influenza pandemics." – This is a huge jump of reductionism. It's highly unlikely that we will be able to predict future influenza pandemics even after knowing all the evolutionary constraints of its genome.

Line 53: Please, define epistasis.

Line 61: Please define parallel evolution.

Lines 76-84: Authors should keep the hypothesis and predictions in the intro but the methodological details such as the actual RF measure should go to methods section.

Line 425: What is a tanglegram? What substitution model was used to reconstruct phylogenies for each vRNA segment? Same model for all?

Reviewer #3 (Recommendations for the authors):

I have no further specific suggestions or critiques that are not in the other sections.

---

## [Author Response]

Essential revisions:In this study, the authors combined phylogenetics and molecular assays to demonstrate parallel evolution across vRNA segments in influenza A virus subtypes, highlighting the importance of RNA-RNA interaction as a novel driver of viral genomic evolution. While reviewers agree that the work is important and likely to have implications for pathogen evolution and emerging infections, there are several major concerns that have been raised in terms of clarity of working hypotheses, methods and interpretations.To cite a few, for example, the rationale behind (1) leaving out sequences that did not resolve well in the phylogenic analyses; (2) relating RNA segment co-localization to similar phylogenetic patterns was unclear and thus, requires more qualification.Unless all the queries raised by the reviewers are addressed, at present, it is difficult to truly judge on its content and what is supported by the data produced.Reviewer #1 (Recommendations for the authors ):In this paper, authors did a fine job of combining phylogenetics and molecular methods to demonstrate the parallel evolution across vRNA segments in two seasonal influenza A virus subtypes. They first estimated phylogenetic relationships between vRNA segments using Robinson-Foulds distance and identified the possibility of parallel evolution of RNA-RNA interactions driving the genomic assembly. This is indeed an interesting mechanism in addition to the traditional role for proteins for the same. Subsequently, they used molecular biology to validate such RNA-RNA driven interaction by demonstrating co-localization of vRNA segments in infected cells. They also showed that the parallel evolution between vRNA segments might vary across subtypes and virus lineages isolated from distinct host origins. Overall, I find this to be excellent work with major implications for genome evolution of infectious viruses; emergence of new strains with altered genome combination.I am wondering if leaving out sequences (not resolving well) in the phylogenic analysis interferes with the true picture of the proposed associations. What if they reflect the evolutionary intermediates, with important implications for the pathogen evolution which is lost in the analyses?

We fully appreciate this concern and have explored this extensively. One principle assumption underlying the approach we outline in this manuscript is that the trees analyzed are robust and well-resolved. We use tree similarity as a correlate for relationships between genomic segments, so the trees must be robust enough to support our claims, as we have clarified in lines 127-130. We initially set out to examine a broader range of viral isolates in each set of trees, but larger trees containing more isolates consistently failed to be supported by bootstrapping. Bootstrapping is by far the most widely used methodology for demonstrating support for tree nodes. We provided the closest possible example to the trees presented in this manuscript for comparison. We took all 84 H3N2 strains from 2005-2014 analyzed in replicate trees 1-7 and collapsed these sequences into one tree for each vRNA segment. Author response image 1**,** specifically provided for the reviewers**,** illustrates the resultant collapsed PB2 tree, with bootstrap values of 70 or higher shown in red and individual strains coded by cluster and replicate. As expected, the majority of internal nodes on such a tree are largely unsupported by bootstrapping, indicating that relaxing our constraint of 97% sequence identity increases the uncertainty in our trees.

Because we agree with Reviewers #1 and #3 on the critical importance of validating our approach, we determined the distances between these new collapsed trees using a complementary approach, Clustering Information Distances (CID), that is independent of tree size (Figure 2 —figure supplement 2A and Author response image 1 and C). Larger trees containing all sequences yielded pairwise vRNA relationships that are largely similar to those we report in the manuscript (R^2^ = 0.6408; *P* = 3.1E-07; Author response image 1 vs. 1C), including higher tree similarity between PB2 and NA over NS. This observation strengthens the rationale to focus on these segments for molecular validation and correlate parallel evolution to intracellular localization in our manuscript (Figure 7). However, tree distances are generally higher in Author response image 1 than in Author response image 1, which we might expect if poorly supported nodes in larger trees artificially inflate phylogenetic signal. Given the overall similarity between Author response image 1 and 1C, both methods yield largely comparable results. We ultimately relied upon the more robust replicate trees with stronger bootstrap support.

**Author response image 1. sa2fig1:** The impact of collapsing replicate trees into larger trees on tree distance. A. Sequence alignments of H3N2 viruses from 2005-2014 from replicates 1-7 were combined into individual alignments for each vRNA segment and maximum-likelihood trees were constructed. The PB2 tree is shown with tree tips indicating the cluster and replicate Bootstrap values greater than or equal to 70 are shown in red. B. Pairwise mean Clustering Information Distance (CID) determined for replicate trees (reproduced from Supplemental figure 4B) are shown in a heatmap. C. Pairwise CID determined from collapsed trees are shown in a heatmap.

Lines 50-51: Can you please elaborate? I think this might be useful for the reader to better from replicates understand the context. Also, a brief description on functional association between different known fragments might instigate curiosity among the readers from the very beginning. At present, it largely caters to people already familiar with the biology of influenza virus.

We have added additional information to reflect the complexity of intersegmental interactions and the current standing of the field (lines 49-52).

Lines 95-96 Were these strains all swine-origin? More details on these lineages will be useful for the readers.

We have clarified that all strains analyzed were isolated from humans, but were of different lineages (lines 115-120).

Lines 128-132: I think it will be nice to talk about these hypotheses well in advance, may be in the Introduction, with more functional details of viral segments.

We incorporated our hypotheses regarding tree similarity into the existing discussion of epistasis in the Introduction (lines 74-75 and 89-106).

Lines 134-136: Please rephrase this sentence to make it more direct and explain the why. E.g. "… parallel evolution between PB1 and HA is likely to be weaker than that of PB1 and PA".

The text has been modified (lines 164-167).

Lines 222-223: Please include a set of hypotheses to explain you results? Please add a perspective in the discussion on how this contribute might to the pandemic potential of H1N!?.

We have added in our interpretation of the results (lines 266-268) and expanded upon this in the Discussion (lines 403-407).

Lines 287-288: I am wondering how likely is this to be true for H1N1.

We have expanded on this in the Discussion (lines 394-395).

Reviewer #2 (Recommendations for the authors):Abstract: Please define parallel evolution and mention the methods used, without details, at the beginning.

We described methods within word count limitations (lines 18-20 and 24-26). A definition of parallel evolution is provided in the Introduction (lines 70-71).

Line 29: "Genetic variation is ubiquitous in RNA viruses." I find this sentence vague. I would prefer a clear statement about high genetic variation in RNA viruses.

This sentence has been modified (line 30).

Line 38: Typo, concomitant

This has been corrected (line 40).

Line 38: "Thus, the emergence of pandemic strains is marked by a concomitant alteration in the influenza virus genome triggered by new genetic diversity." – This is a very complex sentence; either simplify or break it into two, please.

This sentence has been simplified (lines 39-40).

Line 41: "Genetic mutation is driven by stochastic processes and is therefore difficult to predict (Andino & Domingo, 2015)." – The relevance is not clear. Either remove or provide more context.

This statement has been removed.

Line 50-52: "It is consequently imperative to identify the evolutionary constraints imposed by intersegmental vRNA interactions, as this may help predict future influenza pandemics." – This is a huge jump of reductionism. It's highly unlikely that we will be able to predict future influenza pandemics even after knowing all the evolutionary constraints of its genome.

We modified this sentence to better reflect our meaning (lines 55-57).

Line 53: Please, define epistasis.

We added a definition of epistasis (lines 58-60).

Line 61: Please define parallel evolution.

We added a definition of parallel evolution (lines 70-71).

Lines 76-84: Authors should keep the hypothesis and predictions in the intro but the methodological details such as the actual RF measure should go to methods section.

We believe it is helpful to explain tree distance in the Introduction since we are targeting a wide audience of readers and, therefore, we respectfully disagree with the reviewer on this point. A brief explanation of the theory behind both distance measures in the resubmission (RF and CID) is included in the Introduction to make this article more accessible to audiences without expertise in phylogenetics.

Line 425: What is a tanglegram? What substitution model was used to reconstruct phylogenies for each vRNA segment? Same model for all?

We have added a definition of a tanglegram (lines 450-451). The substitution model is stated on lines 441-442; we found that this model is optimal for all trees tested based on AIC and log likelihood criteria.